# Schooling Fish from a New, Multimodal Sensory Perspective

**DOI:** 10.3390/ani14131984

**Published:** 2024-07-05

**Authors:** Matz Larsson

**Affiliations:** 1Clinical Health Promotion Centre, Lund University, 22100 Lund, Sweden; larsson.matz@gmail.com; 2School of Health and Medical Sciences, Örebro University, 70182 Örebro, Sweden; 3The Heart, Lung and Physiology Clinic, Örebro University Hospital, 70185 Örebro, Sweden

**Keywords:** lateral line organ, electrosensory system, evolution, predator confusion

## Abstract

**Simple Summary:**

How can fish manage to swim together in such amazing formations? (This behavior is called schooling in scientific terms.) Many have suggested a primary role of vision, but I explore this dazzling phenomenon from a new perspective: fish mainly use acoustic signals (sound/water-movements) produced by their own movements to achieve synchronization and for keeping appropriate distance from their neighbors. I also propose that schooling prey fish confuse predators’ hearing and lateral line as well as their electrosensory system. Swimming together causes overlapping and complex acoustic signals and blurred electric images, which give the school significant survival advantages. We are descendants of fish, and schooling fish behavior might give clues about human rhythmic behavior (such as music). When human pairs are walking at the same pace, they manage to by listening to footsteps. Walking at the same pace provides acoustical advantages, e.g., short noise-free intervals between steps, which improves hearing. In dangerous times, it could allow for detection of predators and stalkers! The regular and predictable sound of human bipedal walking may have stimulated the evolution of rhythm and music, but could schooling fish “dancing around” in the ocean have started the process?

**Abstract:**

The *acoustic hypothesis* suggests that schooling can result in several benefits. (1) The acoustic pattern (AP) (pressure waves and other water movements) produced by swimming are likely to serve as signals within fish shoals, communicating useful spatial and temporal information between school members, enabling synchronized locomotion and influencing join, stay or leave decisions and shoal assortment. (2) Schooling is likely to reduce the masking of environmental signals, e.g., by auditory grouping, and fish may achieve windows of silence by simultaneously stopping their movements. (3) A solitary swimming fish produces an uncomplicated AP that will give a nearby predator’s lateral line organ (LLO) excellent information, but, if extra fish join, they will produce increasingly complex and indecipherable APs. (4) Fishes swimming close to one another will also blur the electrosensory system (ESS) of predators. Since predators use multimodal information, and since information from the LLO and the ESS is more important than vision in many situations, schooling fish may acquire increased survival by confusing these sensory systems. The combined effects of such predator confusion and other acoustical benefits may contribute to why schooling became an adaptive success. A model encompassing the complex effects of synchronized group locomotion on LLO and ESS perception might increase the understanding of schooling behavior.

## 1. Introduction

Aristotle (384–322 BC) described schooling behavior in fish [1], but many significant questions remain. For example, how do fish synchronize movements? What is the ultimate evolutionary explanation, i.e., the adaptive benefit(s) of schooling?

This article will summarize definitions/descriptions of schooling and briefly discuss similar behavior in in animals other than fish. The evolution of schooling, and the sensory systems involved, particularly vision, the inner ear, the lateral line and the electrosensory system (ESS) are explored in the context of *how* and *why* fish are schooling. This work includes *pros* and *cons* for the relative importance of vision vs. the octavo-lateralis system (OLS). Multimodal perception in shoal assortment/join leave-or-stay decisions are also discussed. Finally, it is hypothesized that schooling prey fish confuse their predators’ octavo-lateralis system (OLS) as well as electrosensory system (ESS) to give them significant survival advantages.

### 1.1. Descriptions and Definitions

Synchronized behaviors arise when two or more animals perform the same behavioral changes at the same time [2]. An outstanding and often discussed example is schooling fish, but what is the exact meaning of the term? Schooling of fish is a complex biological phenomenon, but, despite a long history of research, a generally accepted definition of schooling behavior is lacking [3].

Kasumyan and Pavlov suggested that, accounting for the main characters of schooling behavior, a fish school can be defined as “a temporary self-organizing grouping of individuals predominantly of the same species, size and physiological status, in which there are no permanent leaders and intra-group dominance, which is characterized by a close relationship of equipotential individuals that form it, showing a *desire* (my italics) for unification, consistency, coordination and uniformity of actions when swimming and responding to external stimuli” [4]. Perhaps the wording above “showing a *desire* for unification…” best expresses the need for further research concerning underlying perceptual and cognitive processes.

### 1.2. Schooling-like Behavior in Other Animals

Evolutionary theories suggest that joining a group increases the organisms’ fitness by reducing the chance of being caught by predators, improving foraging efficiency and reducing energy costs [5,6].

Similar behavior (joining groups) can be seen in a variety of vertebrates, such as penguins, cormorants [7] and dolphins [8].

Bird flocks in the sky display a similar collective behavior as fish schools; in both cases, many individuals generate and interact with complex flows [9]. Larsson [10] hypothesized that both birds and fish may use acoustic signals (from wing flapping, fin movements) generated by locomotion in order to attain synchronized locomotion. Moreover, Larsson suggested that human bipedal locomotion in small groups became synchronized due to acoustical advantages such as hearing stalkers and predators, and that this, in turn, may have stimulated the human evolution of rhythm and complex vocal learning [11,12,13].

Dolphins display simple up and down fluke motion, and, like human walking, this produces a simple isochronous beat; the authors speculated that the synchronous behaviors by dolphins in their study may have developed in a similar way as synchronous locomotion developed in humans [8]. Interestingly, cetaceans (together with humans, pinnipeds, elephants and bats) are among the few mammal groups with the strongest evidence of complex vocal learning [14]. Larsson and Abbott proposed that a simple isochronous beat produced during group locomotion may be a prerequisite in the evolution of complex vocal learning, and speculated further that acoustic benefits of locomotor–respiratory coordination in fish ancestors may have selected genetic factors and brain circuitry predisposing tetrapod lines, to synchronized movement, vocalization, and vocal learning [12]. However, this article focuses on fish behavior and two fundamental questions: *how* and *why* are fish schooling?

## 2. Schooling Fish

### 2.1. The Evolution of Schooling

Information about the evolution of schooling is rare due to the lack of archeological data. Scientists have extremely limited material that can be used to analyze the emergence and evolutionary transformations of schooling [4,15]. A very rare and impressive example is the finding of a fossil of a complete school of the juveniles of *Erismatotopterus levatus* [16]. This represents, in principle, a “frozen behavior” [4,15] and shows that almost all fish (257 out of 259) are oriented in the same way, with a close distance between the nearest partners, and the school resembles a moving (elongated) polarized school characteristic of modern fish [4,15]. At least a quarter of all species of Teleostei display schooling behavior [4,15]. Teleostei probably appeared approximately 200–220 million years ago, in the middle or at the end of the Triassic period [15,17].

### 2.2. How Do Fish School?

Fish in a school are disposed to keep a certain distance from their neighbors in the actual condition (predator threat, number of fishes, oxygen level, etc.), e.g., schooling minnows, saithe, cod and herring reside in a water volume that is roughly one body length cubed, which means that neighbors on average are about 70% of a body length away [18]. But this gap varies with swimming speed and grade of fish agitation [18,19]. Furthermore, fish commonly choose to shoal or school together with individuals of similar sizes, and mackerel and herring usually choose to school with neighbors with a size within 15% of their own [20]. Fish in schools display no long-term contacts; according to Pitcher and Parrish, there is only a limited set of responses to the movements of the nearest partners, and these reactions seems to be unconditioned reflexes, facilitating the mutual orientation of fish and the unity of an entire school [21].

### 2.3. Sensory Systems Involved

#### 2.3.1. Vision

Vision has often been proposed to govern schooling behavior of fish [3,4,22]. Kasumyan and Pavlov suggested that vision plays a leading role in upholding schooling swimming; for example, vision allows detection of other fish and evaluation of their exterior characters, their orientation, their movements, the direction and other characteristics of swimming, as well as quantitative characteristics of the shoal [4]. A possible remark against this observation is that a fish has few opportunities to visually compare its own size to other fish. Individuals of the sighted form of Mexican tetra inhabit terrestrial water bodies and lead a schooling lifestyle, while blind cave Mexican tetra with completely degenerated eyes stay alone and do not form schools or shoals [23]. Kasumyan et al. meant that this implies that vision (also in general) is the primary sense in schooling behavior [4]; however, an alternative to that may be that few or no predatory fish are present in a cave, diminishing the benefits of schooling behavior.

#### 2.3.2. The Octavo-Lateralis System (Lateral Line and Inner Ear)

The LLO and the inner ear have several overlapping functions, so numerous principles relating to perception and masking are similar [24,25]. Therefore, this article, in many cases, does not differentiate between water movement and sounds that can stimulate the lateral line and the inner ear.

The locomotion of a solitary swimming fish produces relatively simple acoustic signals (Figure 1), but, if the individual joins a group, the individual’s signals will overlap with others, creating complexity that increases with the group size.

Kasumyan and Pavlov also mention that there are examples of obligate schooling fish lacking lateral line canals such as (Clupeidae) [4,27]. However, Clupeidae have inner ear perception that has many overlapping functions with the LLO [25,28], possibly providing similar sensory information as the LLO in schooling.

Though Kasumyan and Pavlov [4] favored a primary role of vision, they also remarked that “the lateral line can facilitate precise control of the location of nearby schoolmates and allow fish to make quick and well-coordinated maneuvers”, and Larsson has proposed an even more multimodal model of perception, including an influential, or even primary, role of inner ear and lateral line perception [29,30,31]. Despite the loss of vision, the aforementioned cave fish, *A. mexicanusis*, is perfectly capable of orienting and detecting and avoiding obstacles by using its lateral line system [32]. During the gliding phase, the flow field produced by the fish around its body is fixed and expectable and the fish count on this for actively sensing their environment with the LLO [32]. Satou et al. suggested that oscillatory stimuli generated by body vibrations may provide important communication signals during social behavior [33,34].

Water permits approximately five times faster sound propagation than air, as well as subtle hydrodynamic imaging [35]. The physical distribution of fields are well-characterized in theory, but less is recognized regarding the manner in which animals use them in complex tasks and under noisy situations [35,36]. Moreover, sensory perception of aquatic animals has primarily been investigated one modality at a time [35,37], so concepts about multiple interactions of different senses are still, to a large extent, speculative [35].

#### 2.3.3. Fact Sheet

The LLO is a superficial sensory system in fish and other aquatic vertebrates consisting of receptors (neuromasts) that can perceive water displacement. Adult fish usually have two forms of neuromasts:neuromasts within the lateral line canal;the superficial, or free, neuromasts in the epithelium of the head, trunk and caudal fin.

The neuromasts contain axonless mechanosensory hair cells alike those seen in the inner ear. They have kinocilium and a polarized bundle of linked microvilli, which decrease in height with increasing distance from the kinocilium [38]. The LLO can perceive close water movement, low-frequency vibrations and liquid currents, including movements of a travelling sound source. The name ‘lateral line’ originates from the bilateral line observed on the trunk of many fish, though, as said, neuromasts may often be located on the head or elsewhere [39].

The anatomy of the peripheral lateral line system varies greatly across species [32]. The distance to source over which the LLO responds is one to two body lengths [28]. Schooling individuals usually move closer than this.

The distance over which the inner ear responds is greater [40].

The ultrastructure, development and phylogeny of the LLO hair cells are similar with those of the inner ear; therefore, the inner ear and the LLO organs are frequently grouped together with the term octavo-lateralis system (OLS) [28].

### 2.4. Smell

In comparison to acoustic transmission, odor dispersal remains more intelligible due to stratification of smelling substances in water [37]. Smell, and how it interacts with other sensory modalities, is comprehensively explored by Atema [37].

### 2.5. Electrosense

Water propagates electric fields, including electromagnetic induction [35]. Notably, evolutionary modifications of the LLO resulted in the development of the electrosensory system (ESS) [41], and Moller has suggested that the ESS is likely to play a role in schooling; for example, in some weakly electric fish (e.g., Marcusenius cyprinoides, Mormyridae), electric signals seem to support group cohesion in turbid water and during migration in the dark [42].

### 2.6. Pros and Cons for the Relative Importance of Vision vs. the OLS

Kasumyan and Pavlov [4] opposed the suggestion of Larsson [30] that schooling arose due to the appearance of a lateral line in fish, arguing that the lateral line was already present in Heterostraci, Thelodonti and Osteostraci, and that these species existed long before the appearance of the first schooling Teleostei [43]. Kasumyan et al. proposed that such a long gap between the time of the appearance of ancestral fish with a lateral line and the arrival of the first schooling fish makes this assumption (lateral line as a driver of evolution) doubtful. But a comment to that may be that lateral line development may not be sufficient because, also, the group members need to be of similar size (Figure 2 and Figure 3) [10], e.g., birds flying in formation are of similar size. Sturgeons and pike are examples of fish that rarely or never school as adults. The possible role of the OLS in the evolution of schooling is comprehensively discussed in [30].

An experiment showed that blinders had a tiny effect on the position of fish relative to their neighbors in the school, while fish with a temporarily disabled lateral line did school in a different way, with fewer correct distance adjustments [45]. Moreover, the progress of schooling behavior in teleost larvae is strictly associated to the lateral line organ (LLO) development [46,47].

Gray and Denton [24] emphasized the importance of the LLO, saying that the almost instantaneous adjustments to swimming direction and speed that characterize schooling are made possible, by detection via the octavo-lateralis system (OLS), of local water pressure changes resulting from the movements of adjacent fish.

According to the above authors, pressure waves produced by slow body actions in water are small and, in this state, vision has an evident advantage. But the relative merits of visual communication vs. sound signals will be reduced when the speed of movements rise, which moreover suggests that sound communication will provide the first clue when quick movements are started in a school [24]. Also, Satou et al. suggested that oscillatory stimuli generated by body vibrations may provide important communication signals during social behavior [33,34].

Schools of juvenile fish are usually of a different quality in composition regarding species and size. They may contain individuals that differ in size by two to three times and belong not to one, but to several species [48]. The observation that schools of juveniles are less stable and easily break up into several schools during quick maneuvering [48] can be explained by the *acoustical hypothesis* since such huge differences in size will result in differences in the acoustic signals produced by locomotion. Notably, as the juveniles grow, the schools become more and more unified in composition and are better able to maintain unity during movements [48]. This unification in size would also reduce differences in the acoustic patterns produced.

### 2.7. Synchronized Locomotion May Boost the Perception of the Sounds of the Surroundings

Fish moving in synchrony will, as a consequence, also be able to discontinue movements concurrently, thereby producing silent interludes during which the reception of critical environmental signals will be possible. As mentioned, schooling fish are largely of similar size and species [20], and, consequently, the hydrodynamic noise they produce by their locomotion will be alike in amplitude as well as frequency [31]. Synchronous sound of similar character has a tendency to be grouped together in the brain [28]. Such auditory grouping is likely to facilitate the discrimination of self- and neighbor-produced water movements from critical environmental signals [31]. Hence, water movement noise created by the school, albeit approaching from different directions, may be perceived as a single source, which differs from other sound sources [31]. Fish in schools generally move at a similar speed and in the same direction; hence, motion *relative* to neighbors will be reduced compared to that in a less coordinated group. Thus, schooling could reduce the masking by noise from nearby fish [31].

Since *hearing specialist fish* display superb temporal resolution skills, they are able to precisely process temporal patterns of sounds perceived [49].

Accordingly, schooling fish may profit acoustically in two ways by synchronous movement: first, silent intervals will be created when the fish stop simultaneously, and, secondly, when they move, the similarity of each individual’s locomotion sound is likely to boost auditory grouping and, accordingly, the perception of the sounds of the surroundings. A school will not be able to move in perfect synchrony (e.g., such as human militaries may achieve), and responses of fish swimming at the end usually lag somewhat behind [48]. Does that contradict the idea of auditory grouping of locomotion sound? Probably not, since the most-masking locomotion sound is likely to be produced by the adjacent fish, which are likely to move more alike due to mutual reinforcing mechanisms (see below). Accordingly, a reduced relative synchronization of the fish in distant parts of the school should not reduce the anti-masking benefits of schooling.

### 2.8. Can Fish Use Locomotion Sound as Signals within the School?

Incidental sounds produced during locomotion (ISOL) are likely to be among the most common sounds heard during the life of most vertebrates; however, the impact of ISOL on animal cognition and behavior are scarcely studied [10].

The LLO responds only to sources that are quite close, approximately one to two body lengths away [28]. As said, the distance to the nearest school-neighbor is usually 0.7 body lengths [19], i.e., roughly the length of the LLO, and, moreover, it has been shown experimentally that dipole sources used to simulate another fish could be perceived when the distance to the source was within the length of the LLO [40]. The gap over which the inner ear responds is, in comparison, larger [40].

Several authors have proposed that sound produced as a by-product of locomotion may play a significant role in animal communication [10,50,51,52,53]. Larsson hypothesized that the perception of ISOL may provide birds with potentially useful information during flight, such as the speed, location in three-dimensional space (distance and direction) and the wing-beat frequency of neighbors [10]. It seems possible that schooling fish may extract similar information from their ISOL produced by fin movements and gill-breathing [10]. As mentioned, the aquatic medium permits approximately five times faster sound propagation then air [35]. Information embedded in ISOL will travel in all directions; hence, it might be used in mutual adaptation among neighbors. The distance to a neighbor may be assessed from a stereotypic sound with a stable sound level, such as ISOL or breathing of neighboring individuals. Similarity in size and phenotype may produce more predictable ISOL in fish, facilitating synchronization in the shoal. The relative amplitude will be influenced by distance. Thus, when a complex sound travels through water, its timbre changes since higher frequencies are damped more than lower frequencies. Coleman [54] showed that humans can use change in resonance effectively to estimate distance when a familiar sound is heard, but studies in other vertebrates are lacking.

### 2.9. Shoal Assortment/Join Leave-or-Stay?

The length of body and type of species are the main stimuli influencing if individuals will join or not when fish shoals meet, although the sensory mechanisms behind such quick decisions (effectuated within a few seconds) have not been identified [55,56]. Krause et al. have demonstrated that an active shoal choice took place [55].

As said, Kasumyan et al. suggested that that vision plays a leading role in this [4]; that vision allows detection of other fish, evaluation of their exterior characters, their orientation, their movements, their direction and other characteristics of swimming.

However, that does not preclude a role for the OLS. Fish of comparable form and size will emit similar pressure waves and water movements, while fish of different size and form will produce dissimilar acoustic signals [31]. Acoustic signals emitted by swimming are likely to provide information about size, distance to neighbors, frequency of fin movements and undulating movements.

All investigated vertebrates, including fish, have shown the capacity to discriminate among sounds on the basis of frequency, and temporal patterns of sound are the most important carriers of acoustic information for teleost fishes [28]. This is likely to provide essential acoustic information in join leave-or-stay (JLS) decisions [10,29,31].

The water movements from a swimming goldfish *Carassius auratus* [57], and three other fish species with differing swimming style [58], showed a clear vortex structure that persisted for a minimum of 30 s. The authors proposed that this was putatively appropriate to be perceived by a piscivorous predator at a distance where vision or hearing frequently fail [58].

Hanke and Bleckmann [58] speculated that a predatory fish can obtain information beyond the plain presence of a wake and learn to interpret such flow shapes by using OLS perception. An analogous idea is that this may also provide information about individuals of the same species [58]. Thus, ISOL and water disturbance from fish may provide useful information in making JLS decisions [10,29,31]. This hypothesis is supported from an experiment by Pitcher, Partridge and Wardle [59], in which saithe (Pollachius virens, Gadidae) wearing temporary blinders could join and school with normal fish, while saithe with inactivated OLS in addition to blinders could not.

Fish may use visual cues to join larger groups [22,60], and Engezer et al. found that zebrafish (Danio rerio, Cyprinidae) demonstrate strong color shoaling preferences in experiments using normally colored zebrafish and the mutant, nacre (with reduced pigmentation) [61]. The authors proposed that color preference was learned since wild-type zebrafish reared with nacre siblings preferred to shoal with nacre zebrafish. However, Larsson proposed an alternative, or complementary, mechanism, but the “learned color preference” in that experiment does not automatically oppose a role of the OLS—fish may learn to link neighbor color with identifiable pressure waves and water movements [31]. An exclusive part of vision in JLS decisions may also be challenged by the observation that Atlantic herring (*Clupea harengus*, Clupeidae) and the Atlantic mackerel (*Scomber scombrus*, Scombridae) displayed equal preference for associating with similar-sized conspecifics in varying light conditions (night or day) [20]. Kimbell and Morrell [62] demonstrated a reduced shoal preference by prey-fish associated with increasing turbidity in the surrounding water. They suggested that *visual cues* were reduced due to turbulence, but an alternative (or at least complimentary) explanation would be *reduced OLS perception* of prey occupying turbulent waterbodies. And it deserves mention again that the ESS has been shown to support group cohesion in turbid water and during migration in the dark [42].

It is intra-school fission, rather than fusion, that is vital in creating body length- and phenotypic-based assortment [63]. Krause, Ward, Jackson et al. [64] proposed that swimming speed may be a (passive) mechanism for determining fish shoal assortment. Larsson proposed that intra-school fission could be indirectly associated to OLS perception: since fish differing substantially in proportions may not achieve an appropriate degree of synchronization of movements in the school, that could increase the likelihood of separation, possibly explaining why fission, rather than fusion, affects shoal assortment the most [31]. Fish of dissimilar size with reduced capabilities to navigate and follow group members lose contact with the shoal. This may also be the case for bigger, fast-swimming shoal members. Since this implies a rather passive mechanism, the term “decision” may be misleading.

## 3. Why Do Fish School?

Many hypotheses concerning evolutionary advantages of schooling have been proposed, such as safety in numbers, visual confusion of predators [15,55], reduction of encounters with predators [65], watching for predators [15,18], mating and foraging and, not least, reduction of energy expenditure [66,67]. However, field and laboratory studies have led to conflicting conclusions regarding a possible hydrodynamic function of schooling [9,68].

### Schooling to Avoid Risk

Fish are liable to school in dangerous situations, for example, when they are in open areas of water bodies without landmarks and with few shelters against predators [48]. It is particularly in such situations that fish tend to be arranged uniformly in shoals and show mutual attraction to one another, resulting in reduced likelihood to be attacked by predators [69].

Numerous aspects affect decisions, movements and, conceivably, the synchronization in a school. The list includes avoiding obstacles, minimizing energy expenditure and circumventing (or gaining) a leading position [70]. Fish in the front may gain an increased food intake, but they are also more exposed to predator attacks and they usually consume more energy by swimming first [71,72].

## 4. Predator Confusion

### 4.1. Visual Confusion

It is commonly proposed that schooling results in puzzling visual signals to predators [48] and may even cause visual mimicry of a large fish [73,74].

*The oddity effect hypothesis* states that individuals that contrast phenotypically from the rest of the group are *visually* (my italicization) odd and are preferentially targeted by predators [75,76,77,78].

Movements of organisms in a group distract/confuse a predator that is trying to focus on one prey [79], while a phenotypically odd individual in a group can reduce these effects of predator confusion, as the predator is better able to focus on this individual for target, attack and capture [79,80]. All studies above (and numerous others) seem to have investigated only visual predator confusion. Moreover, Aivaz, Manica, Neuhaus et al. stated that the oddity effect has been assumed more often than it has been tested [75]. According to Aivaz et al., oddity in size versus oddity in color has not been compared in a single study before, and, thus, they were the first to explore this hypothesis. They used a needlefish–zebrafish predator–prey system. Their results suggested that *size-oddity* causes a similar response from predators as *color-oddity* and both can play a role in shaping predator and prey interactions. Notably, their study could not give clues concerning the impact on OLS perception since the prey and predator were separated by a plastic wall [75]. Overall, OLS- and ESS-perception are scarcely explored and seldom discussed in the frame of predator confusion.

### 4.2. Confusion of the Octavo-Lateralis System

Self-generated stimuli (such as from locomotion) can be detrimental for a fish since they allow for detection by predators and may also affect the detection of potentially relevant novel stimuli [32]. Tactics to avoid the generation of self-generated water movements have been observed in certain fish; e.g., black carp, *Mylopharyngodon piceus* (Richardson 1846) (Xenocyprodidae), spend significantly less time moving and display overall shorter total distance of movement when the predatory snakehead, *Channa micropeltes*, is present (Cuvier1831) (Channidae; [81]). Under low-light conditions, the lateral line system plays an important role in prey detection [82]. The maximum range of detection of hydrodynamic images (the acoustic near-field detected by the lateral line and vestibular organs) is 0.4–2 predator body lengths from the source [83]. The distance over which the inner ear responds is greater [40].

Larsson hypothesized that schooling is likely to confuse OLS perception [31], and, especially in the final stages of a predator attack, the LLO is more important than vision [84]. Moreover, it was suggested that, since fin movements of a single prey fish will act more or less as a point-shaped wave source, the fish will produce a gradient by which predators might localize it (Figure 1) [31]. On the other hand, schooling fish will produce overlapping, complex acoustic signals that conceal this gradient, accordingly confusing the lateral line perception of predators when prey fish move close to one another (Figure 4). The complexity is likely to increase significantly with the number and density of fishes [31], yet a large group of fish is likely to produce more readily detectable (ISOL) signals, possibly attracting predators. However, Larsson speculated that, since schooling fish are arranged in a symmetrical fashion, this may result in a wave-source that emits a flat wave-front, possibly simulating the pressure waves of a large animal [31]. If so, this might balance the risk of readily detectable signals (and, notably, to some extent be analogous with the aforementioned idea of visual mimicry of a large fish [73,74]).

### 4.3. The Electrosensory System

Around 16% of fish species possesses the capacity of passive electroreception [85]. This is a phylogenetically widespread sensory modality in fishes and amphibians. Many predators, for example, sharks, use the electrosensory system to detect and distinguish electrical fields of prey when they attack [85].

Electroreception gives the capacity to detect external underwater electric fields with specialized receptors. In *passive* electroreception, the predator uses low-frequency-tuned ampullary electroreceptors to sense microvolt-range bioelectric fields from prey, without the need to generate its own electric field. *Active* electroreception (electrolocation) has been demonstrated only in the teleost lineages Mormyroidea and Gymnotiformes; in these cases, the animal perceives its environment by generating a weak (< 1 V) electric-organ discharge (EOD) and, in this way, may detect distortions in the EOD-associated field using high-frequency-tuned tuberous electroreceptors [85]. Tuberous electroreceptors can also perceive the EODs of neighboring fishes, thereby facilitating electrocommunication. Moreover, numerous other groups of elasmobranchs and teleosts produce weak (< 10 V) or strong (>50 V) EODs that can be used to communicate with conspecifics or to kill prey, but lack the capacity for electrolocation. Approximately 1.5% of fish species possess electric organs. Crampton [85] suggested that future research in electric fishes may provide important clues about animal communication. The review by Crampton [85] is comprehensible and well-updated; however, it does not include the proposal of Larsson [31] that schooling fishes may confuse the OLS and ESS of predators. Thus, it seems worthwhile to remind the reader here about this hypothesis, not least since experiments have shed new light on how the ESS may be used by predators in a multimodal frame (see below) [35].

When electric fishes interpret objects in their active space, they do not use the surface properties of object (such as bats and cetaceans do during echolocation). Instead, they use electrical properties of the organs/material of which the objects are composed [86]. As such, electric fishes, in essence, can see through these objects [35]. Objects with different impedance than the water cause perturbations of the pattern detected by the electroreceptors (Figure 5). Electrolocated objects cast a form of electric image on the predator. The image increases in size and becomes fuzzier at the edges as distance to the object increases [86]. The ratio of fuzziness over amplitude can be used by the fish to sense distance [86]. However, [87] showed that objects, or, in this particular case, individual prey, have to be approximately five body widths apart to produce distinct signals. If objects are closer, they will form a blurred image. Based on that, Larsson [31] proposed that schooling fish, where individuals usually are much closer than five body lengths, will have the potential to confuse the ESS of predators.

Gardiner, Atema, Hueter et al. [35] investigated the multisensory guidance of a complex behavioral task in in three species of sharks. They tested five senses experimentally in five different phases of hunting behavior, and below is a brief summary of their report:

1. The blacktip shark, *Carcharhinus limbatus*. Coming from below, this shark detects the existence of prey using olfaction (O) and, in daylight, tracks the bulk flow upstream by using O plus Vision (V) or O plus LLO. Seeing the prey, it switches to V to orient and strike from a distance of roughly 2 m. Close to the prey, the strike is adjusted using the LLO. At that time, it switches to ESS to ram-capture the prey (ram feeding = the predator moves forward with its mouth open, engulfing the prey along with the water surrounding it). When the LLO was blocked, it often missed the prey; successful captures involved increased ram. If EES was blocked, it could capture prey using tactile information (T), but, if T was blocked, it would miss. When approaching prey from downstream in the dark, it detected prey by O and tracked the prey using O plus LLO until the prey was at close range (~20 cm), then it oriented and stroked using LLO (using less ram in the capture);

2. The bonnethead, *Sphyrna tiburo*. From downstream, this shark detects prey using O and, in daylight, tracks it using O plus V or O plus LLO; then, it switches to V to orient and strike, but this happens at a closer range (~1 m) than the blacktip shark (at that time, it switches to ESS to capture using ram-biting). When approaching prey from downstream in the dark with V blocked, it detects prey with O and tracks the prey using O plus LLO, but the shark cannot orient or strike and ceases to feed. If V plus LLO is blocked, it detects prey with O, but cannot track and ceases to feed. When approaching prey from upstream with O blocked, it detects prey using V, then orients, strikes and captures. At night (meaning V is blocked), it detects prey with O and tracks it using O plus LLO, but cannot orient and strike and ceases to feed. If ESS is blocked, it misses the prey, even when touching it;

3. The nurse shark, *Ginglymostoma cirratum*. From downstream, this shark detects prey using O, then, during the daytime, tracks prey using O plus V, O plus LLO or O plus T. At a close range, it switches to V, LLO or ESS to orient and strike, then switches to ESS to suction-capture the prey. At night (VV blocked), it detects by O, tracks using O plus LLO or O plus T, orients and strikes (LLO or ESS) as above, but modulates its capture by increasing suction and decreasing ram. When approaching prey from upstream with O blocked, it does not detect the prey and does not feed. If ESS is blocked, it can still capture the prey if it touches it (i.e., like the blacktip shark), but the shark misses the prey if it does not touch it.

The study by Gardiner clearly demonstrated that sharks are multimodal in their perception of prey and that the senses used are shifting due to light, position, blocked senses and other circumstances. Moreover, the study showed how vital ESS and the OLL are, particularly in the final stages of the attacks. In all species, electric fields guide the timing of jaw opening with millisecond precision [35]. Accordingly, if schooling behavior confuses electrolocation and lateral line perception of sharks, schooling is likely to result in increased odds of survival. Not only sharks, but, in total, around 16% of fish species possesses the capacity for passive electroreception [85], and even more fish species have an LLO. Thus, if schooling confuses both LLO- and ESS-perception of predators it seems likely that this combined result could have been a major incentive in their evolution. More studies like the studies by New et al. [84] or Gardiner et al. [35], including other predatory fishes, are warranted to resolve the relative importance of vision, LLO and ESS in the end stage of attacks. 

The information of naturally occurring biologically relevant or irrelevant lateral line stimuli is still very restricted, and measuring natural stimuli in field studies is complicated [32]. Pressure waves can be recorded with hydrophones, but these are usually too big to measure the small-scale pressure changes that are relevant for the LLO system [88]. Hot-wire anemometers or laser Doppler anemometers are better suited to measure these small-scale water motions, but they are fragile and expensive. Additionally, they measure flow velocity only at a single point in space, which is *why* they cannot provide spatial information [32], and tank studies preclude the useful analysis of directional sound [35]. Investigating confusion of predators’ OLS and ESS due to schooling seems to be a complicated task. Accordingly, technical constraints may be an explanation as to why the OLS system and ESS are scarcely investigated in association with schooling and predator confusion.

## 5. Conclusions

Theoretical models predict that predator confusion of OLS and ESS is plausible, but experimental evidence is lacking. One explanation may be that such studies are a lot more complicated than investigations of visual systems. It is easier and tempting to start looking for a missing key in the light of the streetlamp, but now it seems timely to expand the field. Many questions remain about schooling and visual perception, but researchers ought to think also in a more *lateral* direction; in other words, they should include the LLO and ESS in future studies about schooling and predator confusion. This could possibly solve many issues about the ultimate explanation, i.e., the evolutionary background of schooling. A bonus might be new clues about fish descendants, such as birds flying in formation, and possibly also new ideas concerning complex vocal learning.

## Figures and Tables

**Figure 1 animals-14-01984-f001:**
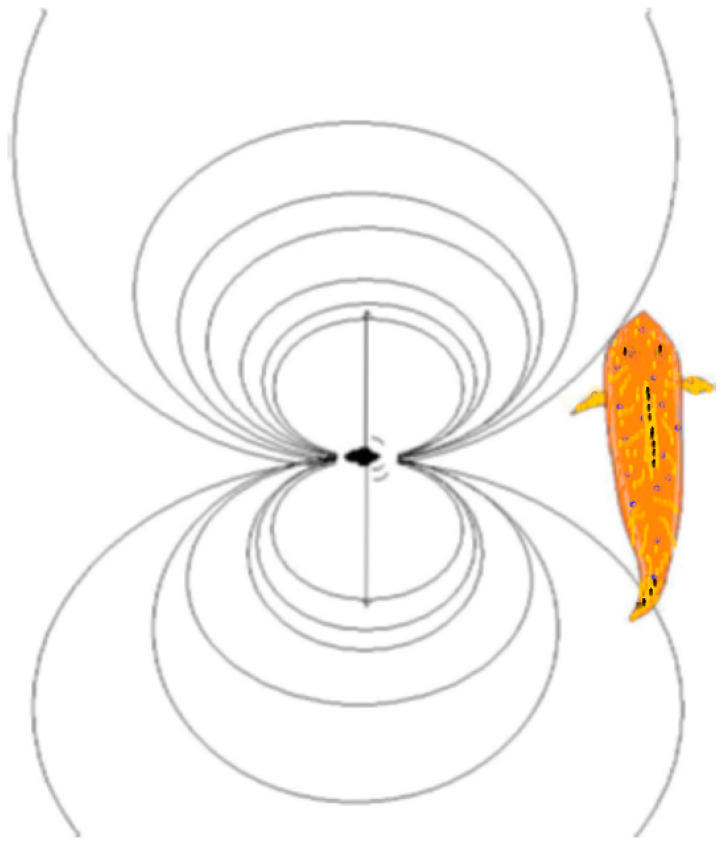
The black spot in the middle is a small fish emitting sound and water movements during locomotion while swimming close to a predator with a lateral line organ. Vibrations, swimming animals, breathing, vocalizations and other mechanical disturbances will generate a steep pressure gradient close to the source, giving rise to a net flow of water. This water flow will obscure particle compressions and rarefactions; as a consequence, near the source, water movements will be more powerful than the propagated pressure wave. These pressure changes are perceived both by the lateral line system and the inner ear. The lateral line, with many densely grouped sampling points, requires a steep spatial gradient for stimulation, but, accordingly, it will be able to resolve that gradient in spatial detail. The auditory system may react to a similar pressure gradient by integrating the differences in pressure along the contralateral sides of its body, but the inner ear cannot resolve spatial details of the stimulus field [25]. Isopressure curves were modelled through dipole flow equations [26].

**Figure 2 animals-14-01984-f002:**
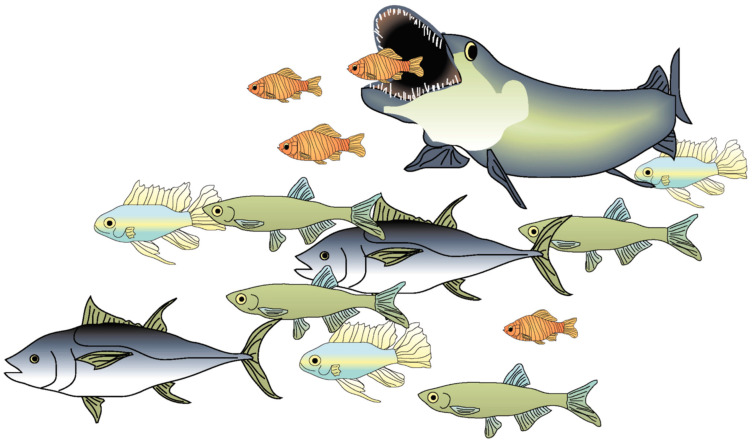
The risk of being eaten by nearby fish may have contributed to evolutionary change towards shoals with individuals of similar size. It is likely that advancement of the OLS was fundamental in the start of predatory behavior [38,44]; moreover (and consequently), this would have increased the risk of being detected and eaten by a larger fish. Conversely, the OLS would have supported small fish to detect and avoid bigger fish. The figure is reproduced from [30] with permission from *Current Zoology*.

**Figure 3 animals-14-01984-f003:**
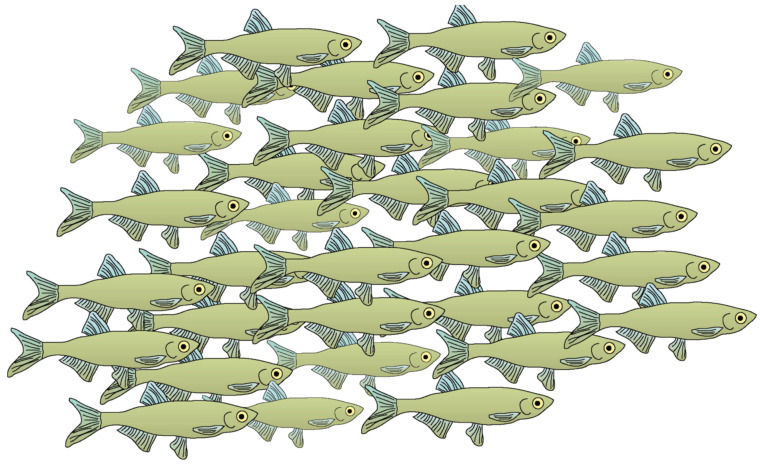
Because of cannibalism, fish may have developed a predisposition to join fish of similar size. Fish of comparable size and body shape will emit similar hydrodynamic signals; such incidental sound of locomotion may be useful in JLS decisions, which may contribute to shoal homogeneity. The figure is reproduced from [30] with permission from *Current Zoology*.

**Figure 4 animals-14-01984-f004:**
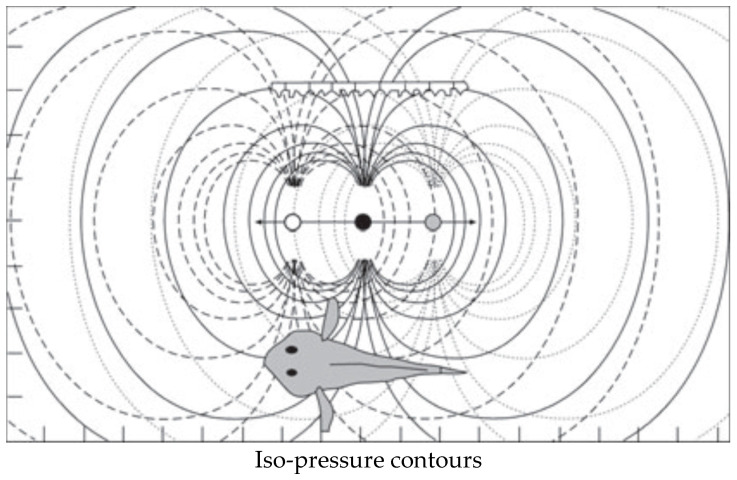
The white, black and gray circles represent small prey fish swimming close to a larger predatory fish (gray). Since the prey fishes are closely situated, the hydrodynamic signals they produce will overlap. Increasing the number and/or reducing the distance between prey fishes will create more overlapping and, thus, more complex signals. This may result in significant predator confusion in regard to the OLS system. The figure is reproduced from [31] with kind permission from *Fish and Fisheries*, Wiley-Blackwell, Hoboken, New Jersey, U.S.

**Figure 5 animals-14-01984-f005:**
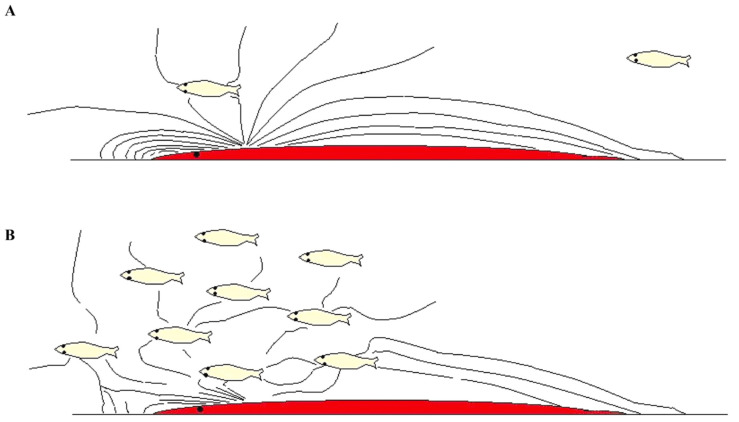
The predatory fish (red) uses the electrosensory system to detect and distinguish electrical fields of prey. (**A**) An isolated fish is easy to detect and localize. (**B**) In a school, the individual fish is difficult to perceive, since the “electrical landscape” is more complicated. Prey fish (yellow) must be about five body widths apart to produce separate signals, or else they will form a blurred image [87]. The figure is reproduced from [30] with permission from *Current Zoology*.

## Data Availability

The original contributions presented in the study are included in the article, further inquiries can be directed to the corresponding author.

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
