# Peer review of "Schooling Fish from a New, Multimodal Sensory Perspective"

_animals, 2024, doi:10.3390/ani14131984_

Round 1
Reviewer 1 Report
Comments and Suggestions for Authors
The author presents a compelling and potentially important review of the literature on our understanding of schooling behavior in fish. She concludes that predators use several sensory systems to detect and capture prey including the poorly studied sensory systems, LLO and ESS. In most fish, schooling is a multi-risk strategy that disrupts signals used by predators to target prey including visual, olfactory, electric fields, LLO, and ESS. The literature review was well organized and readable throughout.
Line 34: "to give them significant survival advantages."
Comments on the Quality of English Language
There were fewer than 20 grammatical issues that were not of sufficient concern to note. The author might wish to have a second party text edit the manuscript to tweak a word here and there.
Author Response
Comments and Suggestions for Authors
The author presents a compelling and potentially important review of the literature on our understanding of schooling behavior in fish. She concludes that predators use several sensory systems to detect and capture prey including the poorly studied sensory systems, LLO and ESS. In most fish, schooling is a multi-risk strategy that disrupts signals used by predators to target prey including visual, olfactory, electric fields, LLO, and ESS. The literature review was well organized and readable throughout.
Line 34: "to give them significant survival advantages."
Response: added survival
Response: Many thanks for encouraging words and an excellent summary of the hypothesis.
Comments on the Quality of English Language
There were fewer than 20 grammatical issues that were not of sufficient concern to note. The author might wish to have a second party text edit the manuscript to tweak a word here and there.
Reviewer 2 Report
Comments and Suggestions for Authors
This paper provides an overview of how cluster fish respond to behavior and patterns through lateral and inductive responses, which has certain research significance and scientific value. But there are also some minor issues, especially in terms of writing style and annotation of atlas references. For example: 1) The exclamation mark !!! after the title of many pictures!!! It should be deleted. I haven't seen this expression before, and the title is too long and not concise enough. 2) Line 365, Exploring the how? - means to find the proximate reasons, while the question why? - means an exploration of the ultimate reason, i.e. the evolutionary background of school- 366 ing fish behavior. This sentence is very colloquial. Do you need to explain these two questions, after all, this is an international journal paper. 3)Suggest streamlining the sentences throughout the entire article, extracting important viewpoints and research conclusions from the article
Comments on the Quality of English LanguageThis paper provides an overview of how cluster fish respond to behavior and patterns through lateral and inductive responses, which has certain research significance and scientific value. But there are also some minor issues, especially in terms of writing style and annotation of atlas references. For example: 1) The exclamation mark !!! after the title of many pictures!!! It should be deleted. I haven't seen this expression before, and the title is too long and not concise enough. 2) Line 365, Exploring the how? - means to find the proximate reasons, while the question why? - means an exploration of the ultimate reason, i.e. the evolutionary background of school- 366 ing fish behavior. This sentence is very colloquial. Do you need to explain these two questions, after all, this is an international journal paper. 3)Suggest streamlining the sentences throughout the entire article, extracting important viewpoints and research conclusions from the article
Author Response
Comments and Suggestions for Authors
This paper provides an overview of how cluster fish respond to behavior and patterns through lateral and inductive responses, which has certain research significance and scientific value. But there are also some minor issues, especially in terms of writing style and annotation of atlas references. For example: 1) The exclamation mark !!! after the title of many pictures!!! It should be deleted. I haven't seen this expression before, and the title is too long and not concise enough.
Response: The language is now checked and adjusted by an expert in American English. The exclamation marks are deleted (they were to remind about permisson to reuse figures).
The title is changed to the more concise : Schooling Fish from a New, Multimodal Sensory perspective
2) Line 365, Exploring the how? - means to find the proximate reasons, while the question why? - means an exploration of the ultimate reason, i.e. the evolutionary background of school- 366 ing fish behavior. This sentence is very colloquial. Do you need to explain these two questions, after all, this is an international journal paper.
Response: Deleted the sentence.
3)Suggest streamlining the sentences throughout the entire article, extracting important viewpoints and research conclusions from the article
Response: I followed the advice of the Sectional Editor to give a brief summary of the article's content in the Introduction, which I think will fulfil a simular purpose.